# Clinical Impact of Recipient-Derived Isoagglutinin Levels in ABO-Incompatible Hematopoietic Stem Cell Transplantation

**DOI:** 10.3390/jcm12020458

**Published:** 2023-01-06

**Authors:** Minjeong Nam, Mina Hur, Hanah Kim, Tae-Hwan Lee, Gun-Hyuk Lee, Sumi Yoon, Seungman Park, Sung Yong Kim, Mark Hong Lee

**Affiliations:** 1Department of Laboratory Medicine, Korea University Anam Hospital, Seoul 02841, Republic of Korea; 2Department of Laboratory Medicine, Konkuk University School of Medicine, Seoul 05030, Republic of Korea; 3Department of Laboratory Medicine, Chung-Ang University College of Medicine, Seoul 06973, Republic of Korea; 4Department of Laboratory Medicine, National Cancer Center, Seoul 10408, Republic of Korea; 5Division of Hematology-Oncology, Department of Internal Medicine, Konkuk University School of Medicine, Seoul 05030, Republic of Korea

**Keywords:** recipient-derived isoagglutinin, hematopoietic stem cell transplantation, survival, graft-versus-host disease, RBC transfusion, platelet engraftment

## Abstract

ABO incompatibility is not considered a contraindication for hematopoietic stem cell transplantation (HSCT). We hypothesized that recipient-derived isoagglutinin (RDI) levels could play a critical role in clinical outcomes. In this study, we compared clinical outcomes such as survival, GVHD, infection, relapse, transfusion, and engraftment, among ABO-compatible patients (ABOc), ABO-incompatible patients (ABOi) with low RDI, and ABOi patients with high RDI. The ABOi with high RDI group was defined as recipients with more than 1:16 RDI levels. We analyzed 103 recipients (ABOc, 53; ABOi with low RDI, 36; ABOi with high RDI, 14). The ABOi with high RDI group showed a decreased 1-year survival and increased acute GVHD grade IV and RBC transfusion (*p* = 0.017, 0.027, and 0.032, respectively). The ABOi with high RDI group was an independent risk factor for increased death, RBC transfusion, and poor platelet (PLT) engraftment (odds ratio (OR) = 3.20, *p* = 0.01; OR = 8.28, *p* = 0.02; OR = 0.18, *p* = 0.03, respectively). The ABOi with high RDI group showed significantly delayed PLT engraftment. In conclusion, this is the first study underscoring high RDI levels as a marker predicting unfavorable outcomes in ABOi HSCT.

## 1. Introduction

Allogeneic hematopoietic stem cell transplantation (HSCT) is a curable modality for hematological or non-hematological malignancies. For HSCT, matching human leukocyte antigens is the most critical part of an appropriate donor selection process [1]. The ABO blood group is expressed throughout the bodily system, such as in red blood cells (RBCs), platelets (PLTs), the organ endothelium, and even in plasma, though not in pluripotent or hematopoietic progenitor cells. ABO incompatibility is not considered the main barrier to transplantation [2]. Among recipients undergoing HSCT, transplantation procedures for approximately 40–50% of recipients are performed across the ABO blood group barrier [3]. ABO-incompatible (ABOi) HSCT can be divided into three groups, according to the blood types of the recipient and the donor: (1) major incompatibility, which is characterized by the presence of recipient antibodies against donor ABO antigens; (2) minor incompatibility, which is characterized by the passive transfer of donor antibodies to the recipient; and (3) bidirectional incompatibility, which is characterized by the combination of major and minor incompatibility [4].

Although not as critical as HLA mismatching, the clinical outcomes in ABOi HSCT are generally considered worse than those in ABO-compatible (ABOc) HSCT, with adverse effects on survival, graft-versus-host disease (GVHD), relapse, and the engraftment of neutrophils and PLT [4,5,6,7,8,9,10,11,12]. Incompatible antibodies against donor RBC antigens were correlated with the risk of unfavorable outcomes after ABOi HSCT [13]. In addition, ABO incompatibility between donors and recipients can cause a hemolytic reaction and delayed RBC recovery due to the reaction of recipient antibodies against donor ABO antigens [14]. Therefore, before undergoing ABOi HSCT, it is a routine clinical practice to remove antibodies or incompatible RBCs from the donor or remove antibodies against donor RBC or residual RBCs from the recipient [15]. These antibodies directed against ABO antigens are known as isoagglutinin [2].

Theoretically, recipient-derived isoagglutinin (RDI) levels are reduced through a conditioning regimen before HSCT [16]. Engrafted donor B cells progressively produce donor antibodies and replace RDI with donor-derived isoagglutinin (DDI). The DDI and RDI kinetics can occur in different forms, depending on the type of conditioning regimen, the stem cell source, and HLA mismatching [17]. In several previous studies, high DDI levels in HSCT recipients were associated with significantly high levels of hemolytic anemia [17,18]. In addition, DDI levels showed a strong correlation with acute GVHD, and DDI levels could be a valuable marker used to predict the incidence of acute GVHD after ABOi HSCT [13,19]. Only one study on RDI levels in ABOi HSCT was conducted, showing that RDI levels decayed more rapidly in recipients with acute GVHD than in recipients without acute GVHD [20]. However, no studies have analyzed the potential association between pre-existing RDI levels and clinical outcomes after HSCT.

Pre-existing RDI levels before ABOi HSCT can be directly related to immunohematological complications and unfavorable outcomes. Measurements of pre-existing RDI levels can provide valuable information on the clinical decisions regarding HSCT. In this study, we aimed to investigate the associations between pre-existing RDI levels and clinical outcomes. We hypothesized that pre-existing RDI levels could play a critical role in survival, GVHD, infection, relapse, transfusion, and engraftment among ABOc patients, ABOi patients with low RDI, and ABOi patients with high RDI.

## 2. Materials and Methods

### 2.1. Study Population

In this study, we enrolled a total of 138 HSCT recipients at the Konkuk University Medical Center (KUMC), Seoul, Korea. All recipients underwent allogeneic HSCT with hematological or non-hematological malignancies during the period from December 2006 to December 2021. The median follow-up period of all recipients was 593 days or approximately 1.6 years (interquartile range (IQR), 198–2536 days; range, 11–5671 days); the median period for deceased recipients was 209 days (IQR 139–436 days; range 11–3553 days) and that of surviving recipients was 2525 days (IQR 1251–4009 days; range 297–5671 days). The final follow-up date for surviving recipients was determined to be 31 July 2022. Recipients were excluded if they (1) underwent autologous HSCT, (2) underwent HSCT twice, or (3) were pediatric patients younger than 18 years old. We analyzed 103 recipients with HSCT (ABOc, 53 recipients; ABOi, 50 recipients). We divided ABOi recipients into two groups according to RDI levels: the ABOi with low RDI group (36 recipients) and the ABOi with high RDI group (14 recipients). The ABOi with low RDI group was defined as recipients with RDI levels of 1:1 to 1:8, and the ABOi with high RDI group was defined as recipients with RDI levels of more than 1:16. All RDI levels were transformed via the log base 2 function for statistical analyses. Demographic and clinical data, including age, sex, diagnosis, ABO incompatibility, conditioning regimen, incidences of infection, GVHD, relapse, CBC count, and transfusion information, were collected from electronic medical records. The characteristics of recipients and donors are summarized in Table 1. The institutional Review Board of the KUMC reviewed this study protocol and exempted the approval of the study with waived informed consent (IRB 2020-09-026).

### 2.2. Measurement of Isoagglutinin Levels

Recipient samples were collected and measured before HSCT. RDI levels were measured manually by means of the immediate spin method. First, 0.2 mL of normal saline was aliquoted into ten plain tubes, and 0.2 mL of recipient serum was mixed into the first tube with label 1. The mixture was serially diluted two-fold from 1:1 to 1:1024. Then, 0.2 mL of 3% Affirmagen A1 cells or B cells (Ortho Clinical Diagnostics, Raritan, NJ, USA) were added into all tubes, and the mixtures were centrifuged for 15 s at 1000× *g*. The last tube with 1^+^ strength agglutination was determined as the RDI level.

### 2.3. Definition of Clinical Outcomes

The primary outcomes of the study were survival, acute and chronic GVHD, cytomegalovirus (CMV) infection, Epstein–Barr virus (EBV) infection, and relapse. Acute and chronic GVHD was diagnosed and graded using the modified Glucksberg system and the 2014 National Institutes of Health Chronic GVHD Diagnosis and Staging Consensus Recommendations [21,22]. CMV infection was defined by detecting viral protein or CMV DNA in plasma or serum [23]. EBV infection was defined when viral protein or EBV DNA was detected [24]. Relapse was defined via bone marrow examinations and short tandem repeat-based techniques. One-year survival, 5-year survival, and 10-year survival were calculated from the date of HSCT to the date of the last follow-up or death by any cause. The last follow-up of the survivors was censored at 365 days for 1-year survival, at 1825 days for 5-year survival, and at 3650 days for 10-year survival.

The second outcomes of the study were transfusion and engraftment. Transfusion was measured for the requirement of Hb and PLT transfusion. Neutrophil engraftment was defined as the first day when the absolute neutrophil count was more than 0.5 × 10^9^/L for 3 consecutive days. PLT engraftment was defined as the first day when the PLT count was more than 20 × 10^9^/L for 7 consecutive days without PLT transfusion.

### 2.4. Statistical Analysis

All data were checked for normal distribution and homogenous variation by the Shapiro–Wilk test. Data were presented as numbers (percentage) or median values (IQR). The incidences of death, acute and chronic GVHD, relapse, infection, the requirement of transfusion, and engraftment were compared among the ABOc group, the ABOi with low RDI group, and the ABOi with high RDI group using crosstab tables with chi-squared statistics or Fisher’s exact test, depending on the type of data. Factors that significantly affected survival and acute GVHD were evaluated using the Cox proportional hazards model. Multivariate logistic regression analyses were used to identify risk factors for RBC transfusion and PLT engraftment—recipient age, female recipients, myeloablative regimen, acute GVHD, RBC transfusion, ABO incompatibility, donor age, and female donors. The odds ratio (OR) was reported with a 95% confidence interval. The probabilities of 1-year survival in the ABOc group, the ABOi with low RDI group, and the ABOi with high RDI group were calculated using the Kaplan–Meier method. Log-rank tests were employed to compare survival curves among groups. The changes in median levels of Hb, white blood cells (WBCs), and PLT after HSCT were calculated by subtracting the initial levels of Hb, WBC, and PLT at the time of HSCT from those levels at 14, 28, 90, 180, and 365 days from HSCT. For statistical analyses, all data were analyzed using MedCalc Software (version 20.014, MedCalc Software, Ostend, Belgium). All data were rounded to 2–3 effective digits [25]. *p*-values < 0.05 were considered statistically significant.

## 3. Results

The ABOi with high RDI group showed a significantly lower incidence of 1-year survival after HSCT than the ABOi with low RDI group (28.6% vs. 72.2%, *p* = 0.02) (Table 2). Among ten deaths in the ABOi with high RDI group at 1-year follow-up, the causes of death were pneumonia (4/10, 40%), relapse (3/10, 30%), GVHD (2/10, 20%), and organ failure (1/10, 10%). The ABOi with high RDI group showed increased incidences of acute GVHD grade IV compared to the ABOi with low RDI group (21.4% vs. 2.8%, *p* = 0.03). The ABOi with high RDI group had required more RBC transfusions (30 days) after HSCT than the ABOi with low RDI group (78.6% vs. 55.6%, *p* = 0.03). The ABOc group and the low RDI group showed comparable clinical outcomes such as survival, acute GVHD, relapse, infection, transfusion, and engraftment (all *p* > 0.05).

The 1-year survival rates were lower in the ABOi with high RDI group than in ABOc group and ABOi with low RDI group (log rank *p* = 0.003) (Figure 1). At 1-year follow-up, the median survival days were 141 days for the ABOi with high RDI group and 365 days for the ABOc group and the ABOi with low RDI group.

In the multivariate analyses, the ABOi with high RDI group was found to be a detrimental factor for death (OR = 3.20, *p* = 0.01) but not for acute GVHD grade IV (Table 3). The ABOi with high RDI group showed a risk for a high requirement of RBC transfusions and poor PLT engraftment (RBC, OR = 8.28, *p* = 0.02; PLT, OR = 0.18, *p* = 0.03) (Table 4).

In comparison to the ABOc group and the ABOi with low RDI group, the ABOi with high RDI group showed delays in positive changes of up to 180 days and 90 days for WBC and PLT, respectively. Hb level changes among the ABOc group, the ABOi with low RDI group, and the ABOi with high RDI group did not show statistical significance (Figure 2). Hb, WBC, and PLT level changes were comparable between the ABOc group and the low RDI group.

## 4. Discussion

To the best of our knowledge, this is the first study demonstrating an association between pre-existing RDI levels and clinical outcomes in ABOi HSCT through the use of data for more than 15 years in a single center. The ABOi with high RDI group showed lower 1-year survival, increased acute GVHD grade IV and RBC transfusion rates, and delayed PLT engraftment. In contrast, the ABOc group and the ABOi with low RDI group showed comparable clinical outcomes. The results of this study suggested that high RDI levels can be predictive for unfavorable outcomes in terms of survival, acute GVHD, RBC transfusion, and PLT engraftment.

No studies have been conducted on RDI levels, and several studies on the clinical outcomes of ABOi HSCT have shown conflicting results for overall survival (OS). It has been reported that major ABOi HSCT showed a lower OS than minor ABOi HSCT, and that minor ABOi HSCT showed a lower OS than ABOc HSCT [9,26]. In contrast, there have been reports that ABOi HSCT was associated with superior OS and disease-free survival [27,28]. An important finding of this study was related to short-term outcomes, with the ABOi with high RDI group showing a significantly lower 1-year survival rate than the ABOi with low RDI group (28.6% vs. 72.2% *p* = 0.02). The poor survival after HSCT could be affected by the relapse rates and/or the incidence of complications such as GVHD, infection, and delayed engraftment [29]. Considering that pneumonia was the most common cause of death in the ABOi with high RDI group, it can be assumed that infection resulting from poor or delayed engraftment may affect unfavorable clinical outcomes, not EBV or CMV infection.

It is well known that GVHD is mediated mainly by donor T lymphocytes against recipient antigens [30]. ABO antigens are expressed on RBC and endothelial cells and are distributed in tissues throughout the whole body [2]. Thus, donor antibodies may bind to and damage the recipient’s tissue, potentially triggering GVHD [31]. However, studies of the effect of ABO incompatibility on GVHD have shown conflicting results, demonstrating no impact, mild impact, or moderate to severe impacts of donor antibodies on GVHD [6,8,9,10]. This study indicated that although the incidence of acute GVHD grade IV was elevated in the ABOi with high RDI group, statistical significance was not shown in the Cox proportional hazards model. As the pathophysiology of acute GVHD is complex, it can be assumed that the incidence of GVHD grade IV could increase due to multiple factors including high RDI levels, rather than an independent high RDI factor. This would need to be verified through further study.

The well-known significant complications of major ABOi HSCT are RBC hemolysis, delayed RBC engraftment, and pure red cell aplasia, despite plasma exchange or donor RBC depletion [4]. These complications have shown that antibodies produced by the recipients’ immune systems could attack donor RBC antigens. A complication of minor ABOi HSCT is massive hemolysis, with the attacking of recipient RBC antigens by donor antibodies [12]. Similarly to previous studies [5,8], our findings indicated that the ABOi with high RDI group required high levels of RBC transfusion. RBC hemolysis can lead to a high demand for RBC transfusion due to ABO incompatibility.

Previous studies have reported no impact of major ABOi HSCT on PLT or neutrophil engraftment [5,11]. Conflicting with these studies, some studies have shown delayed PLT engraftment in ABOi HSCT, compared to ABOc HSCT (ABOi vs. ABOc: 19.78 days vs. 17.15 days) [6]. ABOi HSCT required more PLT infusions than ABOc HSCT [6]. This study demonstrated that the ABOi with high RDI group showed significantly poor and delayed PLT engraftment compared to the ABOc group and the ABOi with low RDI group. However, the ABOi with high RDI group did not show a statistical difference for PLT transfusion. The medical center where this study was conducted strictly adheres to the transfusion protocols for the management of PLT transfusions. According to the transfusion guidelines, PLT transfusions are only performed when the PLT level is lower than a specific level [32]. Therefore, the ABOi with high RDI group did not show statistical significance for PLT transfusion, regardless of statistically significant delayed PLT engraftment. In addition, the ABOi with high RDI group revealed a slow positive change in WBC counts. This may be explained by the presence of recipient antibodies against the ABO blood group antigen or WBCs. However, previous studies showed conflicting results regarding neutrophil engraftment after major ABOi HSCT due to different study populations and ABO incompatibility [5,6,8,9,11].

Previous studies showed that RDIs exponentially decreased after HSCT, and anti-RDI antibodies occurred at low rates [33,34]. The presence of RDIs that should disappear after HSCT may lead to an unfavorable prognosis [35]. In this study, the higher the initial RDI levels observed before HSCT, the longer it lasted, and this could lead to an unfavorable prognosis. Based on the findings of this study, we cannot suggest a definite mechanism to explain the effect of high RDI levels on the clinical impacts. However, the clinical implication of this study is that RDI can be assessed before HSCT. RDI disappears due to the effect of conditioning chemotherapy and is progressively replaced by DDI. DDIs developed one to three weeks after HSCT, indicating immunohematological reconstitution. According to the results of this study, pre-existing high RDI levels in recipients can predict unfavorable clinical outcomes in ABOi HSCT. Therefore, recipients with high RDI levels can apply an individualized conditioning regimen or consider different stem cell sources.

This study has some limitations. First, this retrospective study was conducted over a long study period to collect sufficient sample numbers. Despite a long follow-up period, we focused on analyzing short-term clinical outcomes, especially regarding transfusion and engraftment. This was because the fate of hematopoietic stem cells could be determined for engraftment for the early period. In this study, the ABOc group, the ABOi with low RDI group, and the ABOi with high RDI group did not show significant differences in terms of 5-year survival, 10-year survival, and chronic GVHD. Second, samples were collected over more than 15 years, but our ability to obtain a satisfactory conclusion was limited due to the retrospective nature of the study design and the fact that it was conducted in a single center. A multicenter study or meta-analysis needs to be conducted in order to draw definitive conclusions.

This study is the first to suggest high RDI levels as a marker explaining the unfavorable outcomes in ABOi HSCT. In terms of survival, acute GVHD, RBC transfusion, and PLT engraftment, the ABOi with high RDI group showed unfavorable outcomes, and the ABOc group and the ABOi with low RDI group showed comparable outcomes. In conclusion, clinical outcomes can be predicted by measuring the RDI levels before HSCT, and intense prophylactic treatment and precision medicine for the recipient can be adopted.

## Figures and Tables

**Figure 1 jcm-12-00458-f001:**
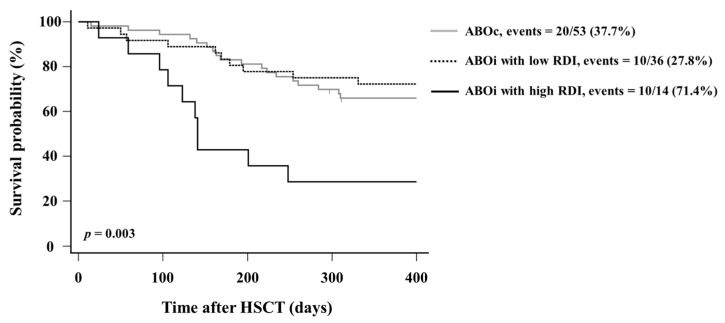
Kaplan–Meier curve for 1-year survival after HSCT. One-year survival rates after HSCT were compared among the ABOc group, the ABOi with low RDI group, and the ABOi with high RDI group. Abbreviations: ABOc, ABO-compatible transplantation; ABOi, ABO-incompatible transplantation; RDI, recipient-derived isoagglutinin; HSCT, hematopoietic stem cell transplantation.

**Figure 2 jcm-12-00458-f002:**
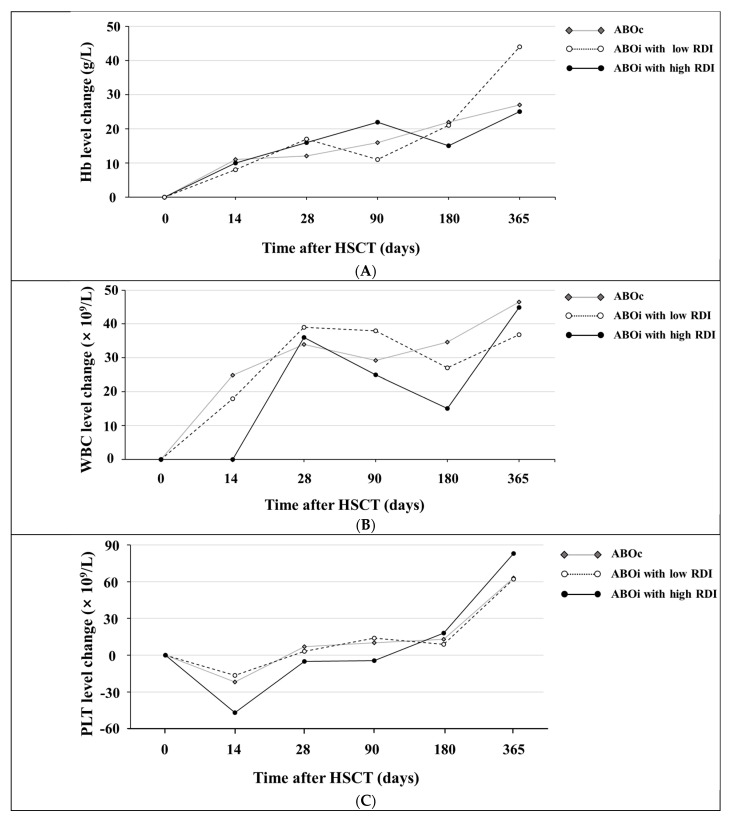
Dynamics of Hb, WBC, and PLT level changes during 1 year after HSCT (ABOc = 53, ABOi with low RDI = 36, ABOi with high RDI = 14). The changes in the levels of (**A**) Hb, (**B**) WBC, and (**C**) PLT represent the results of subtracting the initial value on the day of HSCT from the changed value after 14, 28, 90, 180, and 365 days. The horizontal gray solid line represents the ABOc group, the horizontal dotted black line represents the ABOi with low RDI group, and the horizontal solid black line represents the ABOi with high RDI group. Abbreviations: Hb, hemoglobin; WBC, white blood cell; PLT, platelet; ABOc, ABO-compatible transplantation; ABOi, ABO-incompatible transplantation; RDI, recipient-derived isoagglutinin; HSCT, hematopoietic stem cell transplantation.

**Table 1 jcm-12-00458-t001:** Study population characteristics.

Characteristics	ABOc (*n* = 53)	ABOi (*N* = 50)		*p* Value *
		Low RDI (*n* = 36)	High RDI (*n* = 14)	
Recipient age, years	41.0 (28.8–53.3)	55.5 (51.0–58.0)	38.5 (27.0–51.0)	0.01
Recipient sex				
Male	36 (67.9)	21 (58.3)	10 (71.4)	0.49
Female	17 (32.1)	15 (41.7)	4 (28.6)	0.64
ABO incompatibility				
Major	-	8 (22.2)	9 (64.3)	0.09
Minor	-	21 (58.3)	0 (0.0)	-
Bidirectional	-	7 (19.4)	5 (35.7)	0.54
Diagnosis				
AML	17 (32.1)	13 (36.1)	4 (28.6)	0.79
ALL	12 (22.6)	7 (19.4)	1 (7.1)	0.78
MDS	8 (15.1)	9 (25.0)	4 (28.6)	0.90
Lymphoma	3 (5.7)	2 (5.6)	3 (21.4)	0.67
ALAL	2 (3.8)	2 (5.6)	0 (0.0)	-
PCM	3 (5.7)	0 (0.0)	1 (7.1)	-
CML	4 (7.5)	0 (0.0)	0 (0.0)	-
Other ^†^	4 (7.5)	3 (8.3)	1 (7.1)	0.97
Conditioning regimen				
Myeloablative	24 (45.3)	10 (27.8)	4 (28.6)	0.98
RIC	29 (54.7)	26 (72.2)	10 (71.4)	0.96
Donor age, years	36.0 (27.0–44.5)	35.0 (26.0–46.0)	41.5 (33.0–50.0)	0.25
Donor sex				
Male	31 (58.5)	22 (61.1)	9 (64.3)	0.87
Female	22 (41.5)	14 (38.9)	5 (35.7)	0.90

* *p* values indicate differences between the ABOi with low RDI group and the ABOi with high RDI group. There were no statistically significant differences between the ABOc group and ABOi with low RDI group for all parameters. ^†^ “Other” included 3 aplastic anemia patients, 2 hemophagocytic lymphohistiocytosis patients, and 2 chronic lymphocytic leukemia patients. Abbreviations: ABOi, ABO-incompatible transplantation; ABOc, ABO-compatible transplantation; n, number; RDI, recipient-derived isoagglutinin; AML, acute myeloid leukemia; ALL, acute lymphocytic leukemia; MDS, myelodysplastic syndrome; ALAL, acute leukemia of ambiguous lineage; PCM, plasma cell myeloma; CML, chronic myeloid leukemia; RIC, reduced-intensity conditioning regimen.

**Table 2 jcm-12-00458-t002:** Comparison of clinical outcomes among the ABOc group, the ABOi with low RDI group, and the ABOi with high RDI group.

Outcomes	ABOc (*n* = 53)	ABOi (*N* = 50)		*p* Value *
		Low RDI (*n* = 36)	High RDI (*n* = 14)	
Survival				
1-year survival	33 (62.3)	26 (72.2)	4 (28.6)	0.02
5-year survival	20 (37.7)	13 (36.1)	3 (21.4)	0.52
10-year survival	8 (15.1)	4 (11.1)	3 (21.4)	0.73
aGVHD				
Gr I	8 (15.1)	3 (21.4)	1 (7.1)	0.52
Gr II	5 (9.4)	1 (2.8)	0 (0.0)	0.26
Gr III	4 (7.5)	8 (22.2)	1 (7.1)	0.10
Gr IV	2 (3.8)	1 (2.8)	3 (21.4)	0.03
cGVHD	19 (35.8)	7 (19.4)	1 (7.1)	0.05
Relapse	18 (34.0)	14 (38.9)	4 (28.6)	0.77
Infection				
CMV	18 (34.0)	13 (36.1)	5 (35.7)	0.98
EBV	14 (22.6)	5 (13.9)	0 (0.0)	0.11
Transfusion				
RBC, 30 days	23 (43.4)	20 (55.6)	11 (78.6)	0.03
RBC, 90 days	29 (54.7)	25 (69.4)	12 (85.7)	0.04
PLT, 30 days	16 (30.2)	15 (41.7)	8 (42.9)	0.11
PLT, 90 days	23 (43.4)	18 (50.0)	10 (71.4)	0.17
Engraftment				
Neutrophils at 30 days	26 (49.1)	15 (41.7)	8 (57.1)	0.59
PLT at 90 days	44 (83.0)	30 (83.3)	8 (57.1)	0.08

All data were represented as event numbers (percentages). * *p*-values indicate differences between the ABOi with low RDI group and the ABOi with high RDI group, and there were no statistically significant differences between the ABOc group and the ABOi with low RDI group for all parameters. Abbreviations: ABOc, ABO-compatible transplantation; ABOi, ABO-incompatible transplantation; n, number; RDI, recipient-derived isoagglutinin; aGVHD, acute graft-versus-host disease; gr, grade; cGVHD, chronic graft-versus-host disease; CMV, cytomegalovirus; EBV, Epstein–Barr virus; RBC, red blood cell; PLT, platelet.

**Table 3 jcm-12-00458-t003:** Risk factors for death and aGVHD grade IV, determined using the Cox proportional hazards model.

Factors	Death				aGVHD Grade IV			
	β	SE	*p* Value	OR (95% CI)	β	SE	*p* Value	OR (95% CI)
Age_R	0.00	0.01	0.84	1.00 (0.98–1.03)	0.03	0.05	0.58	1.03 (0.93–1.14)
Female_R	0.83	0.32	0.01	2.28 (1.22–4.26)	0.43	1.34	0.75	1.54 (0.11–21.19)
MR	−0.58	0.31	0.07	0.56 (0.30–1.04)	0.41	1.51	0.79	1.50 (0.08–28.80)
High RDI	1.16	0.43	0.01	3.20 (1.38–7.43)	0.92	1.74	0.60	2.51 (0.08–76.51)
ABOi	−0.03	0.32	0.92	0.97 (0.52–1.81)	−0.62	1.47	0.67	0.54 (0.03–9.54)
Age_D	0.02	0.01	0.08	1.02 (1.00–1.05)	−0.07	0.06	0.24	0.93 (0.83–1.05)
Female_D	0.35	0.31	0.26	1.42 (0.77–2.61)	0.46	1.18	0.69	1.59 (0.16–15.90)

Abbreviations: aGVHD, acute graft-versus-host disease; β, beta coefficient; SE, standard error; OR, odds ratio; CI, confidence interval; R, recipient; MR, myeloablative regimen; RDI, recipient-derived isoagglutinin; ABOi, ABO incompatibility; D, donor.

**Table 4 jcm-12-00458-t004:** Risk factors for RBC transfusion and PLT engraftment, determined using multivariate logistic regression analysis.

Factors	RBC Transfusion				PLT Engraftment			
	β	SE	*p* Value	OR (95% CI)	β	SE	*p* Value	OR (95% CI)
Age_R	−0.01	0.02	0.76	0.99 (0.96–1.03)	0.01	0.02	0.52	1.02 (0.97–1.06)
Female_R	0.84	0.50	0.10	2.31 (0.86–6.16)	−0.74	0.56	0.18	0.48 (0.16–1.42)
MR	−0.16	0.54	0.77	0.85 (0.30–2.46)	−0.50	0.65	0.45	0.61 (0.17–2.19)
High RDI	2.11	0.93	0.02	8.28 (1.33–51.74)	−1.71	0.80	0.03	0.18 (0.04–0.87)
ABOi	0.49	0.49	0.32	1.63 (0.62–4.30)	0.01	0.63	0.99	1.01 (0.29–3.48)
Age_D	−0.06	0.02	0.12	0.94 (0.90–0.99)	0.00	0.03	0.90	1.00 (0.95–1.06)
Female_D	−0.06	0.50	0.91	0.95 (0.35–2.53)	0.09	0.60	0.87	1.10 (0.34–3.54)

Abbreviations: RBC, red blood cell; PLT, platelet; β, beta coefficient; SE, standard error; OR, odds ratio; CI, confidence interval; R, recipient; MR, myeloablative regimen; RDI, recipient-derived isoagglutinin; ABOi, ABO incompatibility; D, donor.

## Data Availability

The data presented in this study are available from the corresponding author upon reasonable request.

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
