# Peer review of "Clinical Impact of Recipient-Derived Isoagglutinin Levels in ABO-Incompatible Hematopoietic Stem Cell Transplantation"

_jcm, 2023, doi:10.3390/jcm12020458_

Round 1
Reviewer 1 Report
In this article, the author described that RDI levels could be a marker for predicting unfavorable outcomes in ABOi HSCT. This work is a retrospective study in a single center, but the long period of follow-up shows high RDI level may be a risk factor, which suggests that patients should take prophylactic treatment before HSCT. However, several issues need to be addressed.
1) The author stated that patients were included between September 2006 to April 2022, which raises a vital question. For some newly included cases, how to get data on 1-year survival, 5-year survival, and 10-year survival? Similarly, how did the author handle those censored data is not found in the text or Figure 1.
2) Although the data show there is a correlation between the high RDI level in ABOi HSCT with lower 1-year survival, increased acute GVHD grade IV and RBC transfusion, and delayed PLT engraftment. However, the weakness is lacking direct links. Additionally, some studies have proved that anti-recipient isohemagglutinins occur at low rates in ABOi HSCT, and isoagglutinin titers showed a quick and exponential fall after transplantation[1, 2]. Thus, the author may provide reasonable or possible mechanisms to explain why ABOi with high RDI group manifested poorer clinical outcomes in the discussion.
[1] ROWLEY S D, LIANG P S, ULZ L. Transplantation of ABO-incompatible bone marrow and peripheral blood stem cell components[J]. Bone marrow transplantation (Basingstoke), 2000,26(7):749-757.
[2] Adkins B D, Andrews J, Sharma D, et al. Low rates of anti-recipient isohemagglutinins in ABO incompatible hematopoietic stem cell transplants[J]. Transfusion and Apheresis Science, 2021,60(1):102965.
3) As engrafted donor B cells progressively produce donor antibodies and replace RDI with DDI. the author also mentioned that DDI levels could be a valuable marker to predict the occurrence of acute GVHD after ABOi HSCT. What are the significances and advantages to detect RDI levels in clinical HSCT.
4) Chi-square test should not be performed on the data set when more than 20% of the cells have an expected frequency of less than 5, or some cells have an expected frequency smaller than 1.
5) Some minor errors should be revised after thoroughly checking. For example, the percentages were miscalculated in the table1:15 (28.3),10 (18.9),4 (7.5).
Author Response
Reviewer #1
Comments to the Author
In this article, the author described that RDI levels could be a marker for predicting unfavorable outcomes in ABOi HSCT. This work is a retrospective study in a single center, but the long period of follow-up shows high RDI level may be a risk factor, which suggests that patients should take prophylactic treatment before HSCT. However, several issues need to be addressed.
- The author stated that patients were included between September 2006 to April 2022, which raises a vital question. For some newly included cases, how to get data on 1-year survival, 5-year survival, and 10-year survival? Similarly, how did the author handle those censored data is not found in the text or Figure 1.
Thank you for your comment. In this study, we modified/added the following sentences in the Method section. According to your comment, Figure 1 has been modified to show the censored data
This study enrolled a total of 138 HSCT recipients at the Konkuk University Med-ical Center (KUMC), Seoul, Korea. All recipients underwent allogeneic HSCT with hematological or non-hematological malignancies from December 2006 to December 2021. The median follow-up period of all recipients was 593 days or approximately 1.6 years (interquartile range [IQR], 198 – 2,536 days; range, 11 – 5,671 days); median levels of deceased recipients, 209 days (IQR 139 – 436 days; range 11 – 3,553 days) and those of surviving recipients, 2,525 days (IQR 1,251 – 4,009 days; range 297 – 5,671 days). The final follow-up date for surviving recipients was locked on July 31, 2022. (page 2, line 85)
1-year survival, 5-year survival, and 10-year survival were calculated from the date of HSCT to the date of the last follow-up or death caused by any cause. The last follow-up of the survivors was censored at 365 days for 1-year survival, at 1,825 days for 5-year survival, and at 3,650 days for 10-year survival. (page 4, line 132)
Figure 1. Kaplan-Meier curve for 1-year survival after HSCT. 1-year survivals after HSCT were compared among ABOc group, ABOi with low RDI group, and ABOi with high RDI group. A horizontal grey solid line represented ABOc group, a horizontal dotted black line represented ABOi with low RDI group, and a horizontal solid black line represented ABOi with high RDI group. (page 5, line 185)
- Although the data show there is a correlation between the high RDI level in ABOi HSCT with lower 1-year survival, increased acute GVHD grade IV and RBC transfusion, and delayed PLT engraftment. However, the weakness is lacking direct links. Additionally, some studies have proved that anti-recipient isohemagglutinins occur at low rates in ABOi HSCT, and isoagglutinin titers showed a quick and exponential fall after transplantation[1, 2]. Thus, the author may provide reasonable or possible mechanisms to explain why ABOi with high RDI group manifested poorer clinical outcomes in the discussion.
(1) ROWLEY S D, LIANG P S, ULZ L. Transplantation of ABO-incompatible bone marrow and peripheral blood stem cell components[J]. Bone marrow transplantation (Basingstoke), 2000,26(7):749-757.
(2) Adkins B D, Andrews J, Sharma D, et al. Low rates of anti-recipient isohemagglutinins in ABO incompatible hematopoietic stem cell transplants[J]. Transfusion and Apheresis Science, 2021,60(1):102965.
Thank you for your valuable comment. We added the following sentences in the Discussion section.
Previous studies investigated that RDIs exponentially decreased after HSCT, and anti-RDI occurred at low rates [33,34]. The presence of RDIs that should disappear after HSCT may affect unfavorable prognosis [35]. In this study, it can be explained that the higher the initial RDI levels performed before HSCT, the longer it lasts and can affect the unfavorable prognosis. This study could not suggest a definite mechanism to explain the effect of high RDI levels on the clinical impacts. However, the clinical implication of this study is that RDI can be assessed before HSCT. RDI disappears due to the effect of conditioning chemotherapy and is progressively replaced by DDI. DDIs were developed one to three weeks after HSCT, indicating immunohematological re-constitution. According to the results of this study, pre-existing high RDI levels in recipients can predict unfavorable clinical outcomes in ABOi HSCT. Therefore, recipients with high RDI levels can apply an individualized conditioning regimen or consider different stem cell sources. (page 9, line 276)
In addition, we newly added the following references.
- Rowley, S.D.; Liang, P.S.; Ulz, L. Transplantation of ABO-incompatible bone marrow and peripheral blood stem cell components. Bone Marrow Transplant. 2000; 26, 749-757, doi: 10.1038/sj.bmt.1702572.
- Adkins, B.D.; Andrews, J.; Sharma, D.; Hughes, C.; Kassim, A.A.; Eichbaum, Q. Low rates of anti-recipient isohemagglu-tinins in ABO incompatible hematopoietic stem cell transplants. Transfus Apher Sci. 2021, 60, 102965, doi: 10.1016/j.transci.2020.102965.
- Malfuson, J.V.; Amor, R.B.; Bonin, P.; Rodet, M.; Boccaccio, C.; Pautas, C.; Kuentz, M.; Cordonnier, C.; Noizat-Pirenne, F.; Maury, S. Impact of nonmyeloablative conditioning regimens on the occurrence of pure red cell after ABO-incompatible allogeneic haematopoietic stem cell transplantation. Vox Sang. 2007; 92, 85-89, doi: 10.1111/j.1423-0410.2006.00865.x.
- As engrafted donor B cells progressively produce donor antibodies and replace RDI with DDI. the author also mentioned that DDI levels could be a valuable marker to predict the occurrence of acute GVHD after ABOi HSCT. What are the significances and advantages to detect RDI levels in clinical HSCT.
Thank you for your comment. According to your comment, we modified/added the following paragraph in the Discussion section.
The pre-existing RDI levels before ABOi HSCT can be directly related to immuno-hematological complications and affect unfavorable outcomes. The pre-existing RDI levels can provide valuable information on the clinical decision for HSCT. In this study, we aimed to investigate the associations between pre-existing RDI levels and clinical outcomes. We hypothesized that pre-existing RDI levels could play a critical role in survival, GVHD, infection, relapse, transfusion, and engraftment among ABOc group, ABOi with low RDI group, and ABOi with high RDI group. (page 2, line 76)
However, the clinical implication of this study is that RDI can be assessed before HSCT. RDI disappears due to the effect of conditioning chemotherapy and is progressively replaced by DDI. DDIs were developed one to three weeks after HSCT, indicating immunohematological reconstitution. According to the results of this study, pre-existing high RDI levels in recipients can predict unfavorable clinical outcomes in ABOi HSCT. Therefore, recipients with high RDI levels can apply an individualized conditioning regimen or consider different stem cell sources. (page 9, line 281)
- Chi-square test should not be performed on the data set when more than 20% of the cells have an expected frequency of less than 5, or some cells have an expected frequency smaller than 1.
Thank you for your comment. According to your comment, we modified the following sentence in the Method section.
The incidences of death, acute and chronic GVHD, relapse, infection, the requirement of transfusion, and engraftment were compared in ABOc group, ABOi with low RDI group, and ABOi with high RDI group using crosstab tables with chi-square statistics or fisher’s exact test depending on the type of data. (page 4, line 143)
- Some minor errors should be revised after thoroughly checking. For example, the percentages were miscalculated in the table1:15 (28.3),10 (18.9),4 (7.5).
Thank you for your comment. According to your comment, we corrected the numbers in Table 1.
Table 1. Study population characteristics.
*P values indicated differences between ABOi with low RDI group and ABOi with high RDI group, and there were no statistically significant differences between the ABOc group and ABOi with low RDI group for all parameters.
†Other included 3 aplastic anemia patients, 2 hemophagocytic lymphohistiocytosis patients, and 2 chronic lymphocytic leukemia patients.

Reviewer 2 Report
This study provides great informations about the possible impact of ABO-directed pre-existing antibodies - just before an ABO-incompatible allogeneic stem cell transplantation.
Next points should imperatively be taken into account:
- the recipient-derived isoagglutinin (RDI) level should be expressed in all the paper as the titre (1:1 to 1:1024), in order to describe the high RDI population with international daily practice most-used units. Indeed, the log2 scales used in the manuscript seems not very explicit.
- in table 1, proportion of patients receiving myeloablative regimen seem to be lower in "high RDI" group. Authors should show the p significancy for this item, which can impact on most short-term studied criteria.
- in results, lines 158 and 159, there is an unappropriate sentence, without any sense: "ABOi with high RDI group showed a median level of 141 days for 1-year survival" ?
- about the 1-year survival criteria: all the message of this manuscript is supported by this criteria, with 10 deaths on 14 patients occuring in the first year. Could the authors give more informations about the cause of these ten deaths ? We would appreciate to clearly evaluate the ten deaths situations, in order to appreciate if ABO incompatiblity could -or not- explain these deaths. If the 3 patients who had a grade IV aGVHD died before day 100, it could be useful to explicit this point. Relapse situations do not seem to explain this 1-year overall survival data. Thus, were the deaths to consider as NRM ?
- multivariate analyses clearly show that nor ABO incompatibility nor high RDI score were signficantly risk factor for grade IV aGVHD. Thus, it seems unappropriate to affirm that :
1/ [high RDI levels can be predistable for unfavourable outcomes in terms .... of acute GVHD] (line 210),
2/ ....by acute GVHD grade IV... in line 221.
Author Response
Reviewer #2
This study provides great information about the possible impact of ABO-directed pre-existing antibodies - just before an ABO-incompatible allogeneic stem cell transplantation.
Next points should imperatively be taken into account:
- The recipient-derived isoagglutinin (RDI) level should be expressed in all the paper as the titre (1:1 to 1:1024), in order to describe the high RDI population with international daily practice most-used units. Indeed, the log2 scales used in the manuscript seems not very explicit.
Thank you for your comment. In the manuscript, we converted all numbers expressed on a log2 scale to isoagglutinin titer.
ABOi with high RDI group was defined as recipients with more than 1:16 RDI levels. (page 1, line 29)
ABOi with low RDI group was defined as recipients with 1:1 to 1:8 RDI levels, and ABOi with high RDI group was defined as recipients with more than 1:16 RDI levels. (page 2, line 96)
- In table 1, proportion of patients receiving myeloablative regimen seem to be lower in "high RDI" group. Authors should show the p significance for this item, which can impact on most short-term studied criteria.
Thank you for your comment. According to your comment, we added p values in Table 1.
Table 1. Study population characteristics.
*P values indicated differences between ABOi with low RDI group and ABOi with high RDI group, and there were no statistically significant differences between the ABOc group and ABOi with low RDI group for all parameters.
†Other included 3 aplastic anemia patients, 2 hemophagocytic lymphohistiocytosis patients, and 2 chronic lymphocytic leukemia patients.
- In results, lines 158 and 159, there is an inappropriate sentence, without any sense: "ABOi with high RDI group showed a median level of 141 days for 1-year survival" ?
Thank you for your comment. We modified the following sentence in the Result section.
The 1-year survival rates were lower in ABOi with high RDI group than in ABOc group and ABOi with low RDI group (log rank p = 0.003) (Figure 1). At 1-year follow-up, the median survival days were 141 days for ABOi with high RDI group and 365 days for ABOc group and ABOi with low RDI group. (page 5, line 182)
- About the 1-year survival criteria: all the message of this manuscript is supported by this criteria, with 10 deaths on 14 patients occurring in the first year. Could the authors give more information about the cause of these ten deaths? We would appreciate to clearly evaluate the ten deaths situations, in order to appreciate if ABO incompatibility could -or not- explain these deaths. If the 3 patients who had a grade IV aGVHD died before day 100, it could be useful to explicit this point. Relapse situations do not seem to explain this 1-year overall survival data. Thus, were the deaths to consider as NRM ? .
Thank you for your valuable comment. According to your comment, we modified/added the following sentences in the Result and Discussion sections. In addition, RRM was 2 cases in ABOi with high RDI group. Overall mortality and NRM did not show significant differences, and we presented the results with overall mortality.
Among ten deaths in ABOi with high RDI group at 1-year follow-up, the causes of death were pneumonia (4/10, 40%), relapse (3/10, 30%), GVHD (2/10, 20%), and organ failure (1/10, 10%) (data not shown). (page 4, line 163)
Considering that pneumonia showed the highest cause of death in the ABOi with high RDI group, it can be assumed that infection resulting from poor or delayed engraftment may affect unfavorable clinical outcomes, not EBV or CMV infection. (page 2, line 238)
- Multivariate analyses clearly show that nor ABO incompatibility nor high RDI score were significantly risk factor for grade IV aGVHD. Thus, it seems inappropriate to affirm that :
1/ [high RDI levels can be predistable for unfavourable outcomes in terms .... of acute GVHD] (line 210),
2/ ....by acute GVHD grade IV... in line 221.
Thank you for your comment. According to the comment, we modified the following sentences in the Discussion section.
This study indicated that although the incidence of acute GVHD grade IV was elevated in ABOi with high RDI group, statistical significance was not shown in the Cox proportional hazards model. As the pathophysiology of acute GVHD is complex, it can be assumed that the incidence of GVHD grade IV could increase due to multiple factors including high RDI levels, rather than an independent high RDI factor. It would need to be verified through further study. (page 8, line 246)

Round 2
Reviewer 1 Report
The authors have addressed all my questions, and the current version is acceptable for publication.